# Influence of Immune System Abnormalities Caused by Maternal Immune Activation in the Postnatal Period

**DOI:** 10.3390/cells12050741

**Published:** 2023-02-25

**Authors:** Yo Shimizu, Hiromi Sakata-Haga, Yutaka Saikawa, Toshihisa Hatta

**Affiliations:** 1Department of Pediatrics, Daido Hospital, Nagoya 457-8511, Japan; 2Department of Anatomy, Kanazawa Medical University, Kahoku 920-0265, Japan; 3Department of Pediatrics, Kanazawa Medical University, Kahoku 920-0265, Japan

**Keywords:** maternal infection, maternal immune activation, immune disorders, immune overreaction, immune response failure, epigenetic modification

## Abstract

The developmental origins of health and disease (DOHaD) indicate that fetal tissues and organs in critical and sensitive periods of development are susceptible to structural and functional changes due to the adverse environment in utero. Maternal immune activation (MIA) is one of the phenomena in DOHaD. Exposure to maternal immune activation is a risk factor for neurodevelopmental disorders, psychosis, cardiovascular diseases, metabolic diseases, and human immune disorders. It has been associated with increased levels of proinflammatory cytokines transferred from mother to fetus in the prenatal period. Abnormal immunity induced by MIA includes immune overreaction or immune response failure in offspring. Immune overreaction is a hypersensitivity response of the immune system to pathogens or allergic factor. Immune response failure could not properly fight off various pathogens. The clinical features in offspring depend on the gestation period, inflammatory magnitude, inflammatory type of MIA in the prenatal period, and exposure to prenatal inflammatory stimulation, which might induce epigenetic modifications in the immune system. An analysis of epigenetic modifications caused by adverse intrauterine environments might allow clinicians to predict the onset of diseases and disorders before or after birth.

## 1. Introduction

Postnatal disorders are significantly influenced by the prenatal environment [1]. During World War II, Barker et al. were the first to report that maternal malnutrition was associated with adverse effects on adult health and resulted in an increased risk of cardiovascular diseases in adulthood [2]. A positive correlation between low birth weight and mortality rate from ischemic heart disease was observed. Based on several investigations, they proposed a hypothesis of the developmental origins of health and disease (DOHaD) [3]. This hypothesis indicates that the fetal tissues and organs experience structural and functional changes during critical and sensitive developmental periods due to adverse environments in utero. The association between the risk factors of the maternal environment, such as exposure to adverse intrauterine factors including infection, allergens, oxidative stress, and medicine and disease development, such as ischemic heart disease, has also been reported [4,5,6,7,8,9]. The postnatal disability caused by adverse environments in the fetuses may result in neonatal or infant death or impact on their health as adults [1,2]. These damages affect not only the individual but multiple organs of the whole body and are inherited by the next generation. Maternal immune activation (MIA) is a DOHaD mechanism. It is a phenomenon in which maternal infections, autoimmune diseases, allergies, asthma, atherosclerosis, malignancy, hyperhomocysteinemia, and alcohol consumption activate maternal immune response [10,11,12,13]. An inflammatory response due to cytokine production during gestation is common in these diseases. According to epidemiologic studies on humans, maternal infection during pregnancy causes various disorders associated with abnormal immunity in the offspring, including type 1 diabetes, allergy diseases, psychosis, and neurodevelopmental abnormalities [7,14,15]. Increased plasma levels of cytokines such as interleukin (IL)-1, IL-6, IL-8, IL-12p40, IL-13, and tumor necrosis factor (TNF)-α, as well as the immune system’s activation in response to inflammatory stimuli in monocyte cell culture, have been linked to abnormal immunity in patients with autism-like disorders [10,16,17]. Thus, MIA induces abnormal immunity in offspring. Previous studies have reported that MIA causes alterations in epigenetics, neurotrophin expression, brain structure and function, gut microbiota, oxidative stress response, and mitochondrial dysfunction in the fetus as well as neurodevelopmental disorders, psychosis, cardiovascular diseases, metabolic diseases, and immune disorders in the offspring (Figure 1) [18,19]. Therefore, MIA is a very critical factor associated with the onset of various diseases in offspring.

This review aims to clarify the relationship between the gestational period of exposure to maternal inflammation induced by MIA, the severity of maternal inflammation, the type of cytokine or signaling pathway stimulation, and postnatal immune dysregulation reactions based on the current literature. A new form of severe acute respiratory syndrome coronavirus 2 (SARS-CoV-2) has been spreading worldwide, and maternal infection’s postnatal effects in gestation are of great concern. Although limited, the current reported maternal coronavirus disease 2019 (COVID-19) infection and its postnatal effects are described. Some mechanisms of diseases and disorders associated with MIA have already been reported. Epigenetic alterations caused by MIA directly affect the immune system and might prospect to applicate the treatment and diagnosis using epigenetic technology. The potential application of epigenetic technology in the future is explained. We also reviewed how genes, prenatal environment, and postnatal environment influence the development of postnatal immune disorders, and the positioning of MIA among them is summarized.

## 2. Immunological Disorders Caused by MIA in Humans

### 2.1. Diseases Associated with an Abnormal Immunity Induced by Maternal Infection in Humans

Genetic factors play a critical role in inducing diseases and disorders, with the prenatal environment being a key factor. In epidemiological studies of humans, maternal infection during pregnancy evoked several diseases associated with abnormal immunity, including type 1 diabetes mellitus, allergic diseases, and neurodevelopmental disorders in offspring. The mechanisms include the autoimmune response in type 1 diabetes mellitus, immune hypersensitivity in allergic diseases, and immune overreaction in neurodevelopmental disorders [7,14,15] (Figure 2). Abnormal immunity originating in the prenatal period continues to influence the tissues and organs after birth. It ultimately results in several diseases, such as allergic diseases caused by exposure to postnatal risk factors in offspring. It is necessary to clarify the mechanism of abnormal immunity induced by MIA in offspring.

### 2.2. Mechanism of Diseases and Disorders with Abnormal Immunity Caused by Maternal Infection

#### 2.2.1. Diabetes Mellitus

In a previous meta-analysis, maternal infection was significantly associated with type 1 diabetes mellitus [15]. Maternal infection is driven by T cell response in the fetus and leads to the development of pathogen-specific T cells implicated in autoimmune response. Moreover, maternal infection leads to the production of antibodies and transmission to the fetus through the placenta [20,21]. These immune system alterations in the prenatal period could produce proinflammatory cytokines after birth, resulting in β-cell dysfunction in offspring. 

#### 2.2.2. Allergic Diseases

Exposure to maternal infection in utero may cause allergic diseases, including asthma, eczema, and hay fever, to develop. Previous research has shown that the complex process underlying asthma, an allergic disease, involves prenatal and postnatal risk factors, including genetic variables and prenatal and postnatal environments, including human rhinoviruses and respiratory syncytial virus infections [7]. The prenatal environments, such as maternal viral or bacterial infections, allergen, tobacco smoke, and air pollution, induce epigenetic alterations associated with airway function, mucosal immune response, systemic immune response, and atopic sensitization in offspring [7]. As a result, immunological hypersensitivity and the development of allergy diseases such as asthma, eczema, and hay fever are caused by genetic factors and prenatal exposure to MIA. Farm exposure during gestation may protect infants against allergic diseases, including asthma, hay fever, and eczema [22]. The frequency of contact with farm environments by the mother during gestation correlates inversely with the incidence of allergic diseases in offspring. Contact with live stocks or drinking fresh milk can lead to exposure to multiple bacteria, endotoxin, and fungi. When compared to neonates who had no maternal farm contact, those with a history of maternal farm contact had weaker immune responses. The interferon (IFN)-γ/IL-13 ratio increased after lipopolysaccharide (LPS) stimulation in the farming mother group, and the immune response tended to be Th1 dominant. Moreover, the number of Th9 cells increased in patients with allergies. IL-9 production decreased after LPS stimulation in cord blood mononuclear cells from neonates exposed to maternal farm contact and may be implicated in the suppression of Th9 function. Alterations in the quality and function of neonatal Tregs induced an imbalance in the immune response, causing a Th1-dominant state and suppressing Th9 cell function and IL-9 production [23].

#### 2.2.3. Neurodevelopmental Disorders and Psychosis

Previous reports have suggested that maternal inflammation may play a role in neurodevelopmental disorders in children, such as autism spectrum disorder, attention deficit hyperactivity disorder, specific learning disorders, communication disorders, intellectual disability, and psychosis such as schizophrenia or anxiety in offspring [18,19,24,25,26,27,28]. The risk of neurodevelopmental diseases may depend on infectious agents, gestation period, and infection site [29]. Numerous investigations on inflammatory disorders in children with autism have revealed that several proinflammatory cytokines were elevated in plasma and whole blood cell mRNA, whereas anti-inflammatory cytokines, including IL-4 or IL-10, were unchanged compared with the control group [16,17,30,31,32,33,34,35,36,37]. Thus, anti-inflammatory cytokines are weakly associated with patients with autism. TNF-α levels correlated positively with autism severity, and decreased expression of immunoregulatory genes is related to TNF-α [34]. Although the mechanism underlying abnormal immunity in neurodevelopmental disorders is still poorly understood, dysregulated peripheral blood cell toll-like receptor response to inflammatory stimulation during the postnatal period has been observed in these patients [38]. After birth, maternal inflammation causes abnormal immunity, such as immune overreaction or autoimmunity. This results in damage to the fetal brain caused by neuron impairment because of the continuous production of proinflammatory cytokines. An increased copy number of long interspersed nuclear element-1 (L1) has been identified in induced pluripotent stem cells derived from neurons in the brains of patients with schizophrenia with a 22q11 deletion. An increased L1 copy number was also found in the offspring of mice exposed to polyriboinosinic-polyribocytidylic acid [poly(I:C)]-induced MIA, which exhibit schizophrenia-like behavior [39]. This suggests that the increase in L1 copy number in neurons induced by environmental factors is involved in schizophrenia susceptibility and pathogenesis [39].

Exposure to maternal inflammation from a viral or bacterial infection and autoinflammatory, autoimmune, and allergic diseases cause neurodevelopmental disorders, allergic diseases, and type 1 diabetes with immunological abnormality in humans. A rodent MIA model using poly(I:C) or LPS during the gestation period affected the immune system, resulting in immune overreaction, suppression of immune response, or inhibition of Th2 immune response, in offspring.

## 3. The Impact of Exposure to SARS-CoV-2 Infections in the Prenatal and Postnatal Period

### 3.1. Possible Relationship between Prenatal Environment and COVID-19 Infection in the Postnatal Period

A new form of SARS-CoV-2 has been spreading worldwide, which has caused many deaths. This disease was named COVID-19 [40] and has a wide range of symptoms, including cold symptoms for mild cases and multiple organ failure for severe cases [41,42]. There are many reports about the risk factors of severe COVID-19. First, postnatal risk factors include aging and comorbidities, including diabetes mellitus, hypertension, obesity, asthma, chronic obstructive pulmonary disease, and chronic kidney disease [43,44,45,46]. Second, there are genetic variants of SARS-CoV-2 entry cytoplasm-related mechanisms and abnormal immunity [47]. Third, epigenetic modifications with clinical severity of COVID-19 are related to virus entry and innate immune systems, such as IFN signaling, angiotensin-converting enzyme-2, inflammasome component absent in melanoma 2, and major histocompatibility complex class IC candidates [47,48]. These epigenetic alterations might be related to prenatal stimulation in the fetus or changes caused by COVID-19 in the postnatal period. The association between epigenetic modifications and the prenatal environments in patients with severe COVID-19 needs to be clarified. Thus, an analysis of the association between epigenome changes and the prenatal environment in various diseases, such as COVID-19, is expected to benefit future clinical practice.

### 3.2. Adverse Effects in Infants Exposed to COVID-19 Infection in the Prenatal Period

#### 3.2.1. Neurodevelopmental Disorders

The mechanism of neurodevelopmental disorders, including autism, schizophrenia, learning disability, and ADHD, remains unclear. However, the relationship between maternal viral and bacterial infections in pregnancy and neurodevelopmental disorders in the postnatal period is well known in epidemiologic studies [18,19]. Recently, fetal developmental disorders due to maternal SARS-CoV-2 infection in pregnancy were a primary concern because the inflammatory cytokines implicated in maternal and placental inflammation damage the fetal brain in the prenatal period. As expected, recent studies have suggested that infants of mothers with SARS-CoV-2 infection during pregnancy develop neurodevelopmental disorders at one year of age [49]. Neurodevelopmental disorders, such as autism, schizophrenia, learning disabilities, and ADHD, are often diagnosed after one year and require long-term follow-up beyond the neonatal period.

Additionally, the dysfunction of organs and tissues other than the brain needs to be examined. There are concerns about the direct effects of COVID-19, such as pneumonia or organ failure, and the impact on future generations. Further epidemiological investigations and mechanistic studies are warranted.

#### 3.2.2. Immune Dysfunction

Recent studies reported immune abnormalities caused by SARS-CoV-2 infection in the prenatal period. The immune abnormalities in newborn babies from maternal SARS-CoV-2 infection are affected by maternal infection status. The babies born from mothers with recent or ongoing infection have high cytokine levels in the serum and increase the percentage of natural killer (NK) and regulatory T cells compared with those born from mothers with recovered or uninfected. However, B cells, CD4^+^ T cells, and CD8^+^ T cells had similar percentages in both groups. The serology test of babies does not show a vertical infection from the mother [50,51]. Therefore, immune abnormalities might be associated not with direct fetal infection but with the immune response of maternal SARS-CoV-2 infection. This study evaluates the immune system at birth and has limited results because no other study performed this examination. Furthermore, there are no reports of long-term alteration in the immune system, such as during infancy, childhood, or adolescence. However, long-term changes in the immune system due to maternal infection are expected along with other infections, and further investigation is crucial.

#### 3.2.3. Endothelial Cell Dysfunction

SARS-CoV-2 infection induces endothelial cell dysfunction because of endothelium-related deleterious molecules, including alterations of oxidative stress, inflammation, glycocalyx damage, thrombosis, and vascular tone. The endothelial cells maintain function involved in circulating blood and various tissues or organs homeostasis. Therefore, maternal SARS-CoV-2 infection in the gestation period leads to adverse effects associated with endothelial cell dysfunction. First, severe cases of maternal SARS-CoV-2 infection progress multiple organ failure due to hypercytokinemia and lead to blood flow disturbance from mother to fetus through the placenta because of circulatory failure by the septic shock of SARS-CoV-2 infection [52,53]. Second, previous studies reported that the histopathology findings of maternal SARS-CoV-2 infection in the third trimester detected both fetal vascular and maternal malperfusion. These findings included inflammation in the placenta without SARS-CoV-2 vertical infection via the placenta [54]. Epidemiological studies indicate that maternal SARS-CoV-2 infection causes preterm birth, low birth weight, and stillbirth [55,56,57]. These adverse influences might be associated with blood flow disturbance caused by endothelial cell dysfunction; however, its status is implicated with complicated factors, such as the direct effects of infection or indirect effects of a respiratory disorder, liver injury, kidney injury, and myocardial injury [53].

## 4. Immune Dysfunction Caused by MIA in Animal Models

### 4.1. Immune Dysfunction Caused by Prenatal Exposure to Poly (I:C)

Numerous studies have revealed that immunological disorders are associated with MIA. According to a previous meta-analysis of the association between MIA and immunological dysfunction, midgestational maternal exposure to poly (I:C) caused immune overreaction from the perinatal period through preweaning without inducing inflammatory stimulation after birth (Figure 2) [58,59,60,61]. IL-1β and TNF-α levels in offspring are slightly elevated in the absence of inflammatory stimuli. Although cytokines such as IL-1 or TNF-α are a more immediate inflammatory response to poly (I:C) than to IL-6, the IL-6 level increased significantly [61]. These results showed that immune hypersensitivity acquired during the prenatal period caused the continuous production of inflammatory cytokines after birth without any postnatal inflammatory stimulation. Anti-inflammatory cytokines, including IL-4 and IL-10, were unchanged, and poly (I:C)-induced MIA might not affect anti-inflammatory cytokine levels in offspring [62]. The mechanisms of immune dysfunction caused by maternal infection are associated with altered cellular stress response, gut microbiota, neurotrophin expression, brain structure and function, and neuroimmune regulation [14,15,63]. MIA induced changes in species and distribution in gut microbiota, inflammatory cytokines, such as TNF-α, and the cerebellum associated with gut microbiota; it also plays a critical role in regulating neuroinflammatory response [62]. Furthermore, mouse or human commensal bacteria, which induce Th17 cell response, cause neurodevelopmental disorders in mice [63]. Therefore, the condition or modification of gut microbiota caused by MIA is associated with immune dysfunction implicated in the Th17 immune response or regulation of the neuroinflammatory response.

Few reports using animal models have examined how offspring exposed to maternal inflammatory stimuli in utero respond when exposed to a second inflammatory stimulus after birth. In a case exposed to poly (I:C) prenatally, the serum IL-6, IL-17, and TNF-α levels increased much higher than those in offspring not exposed to poly (I:C) in vivo during the postnatal period; liver necrosis as an organ injury was also detected in this case [58]. The mRNA expression of the binding immunoglobulin protein and activating transcription factor 4, a major regulator and key transcription factor of the unfolded stress response (UPR), was low compared with controls, suggesting that the UPR defect is induced by prenatal MIA exposure. UPR defects cause an inability for tissue and organ homeostasis maintenance due to excess unfolding proteins, leading to excessive activation of the immune response to inflammatory stimuli [58]. The mechanisms of immune overreaction also include an increase in macrophage 1 polarization, activation of innate immunity, and deficit in T regulatory cells [59,60,64]. Cells derived from offspring exposed to poly (I:C) prenatally exhibited increases in IL-1 and IL-12 levels following LPS stimulation in vitro [59,60] (Table 1). These results suggest that prenatal exposure to inflammation due to MIA is an important event associated with postnatal immune dysfunction. The second attack of inflammatory stimulation after birth enhances the MIA-induced immune overreaction in offspring. Moreover, these second-hit models may be involved in various diseases, such as severe infections or autoinflammatory diseases with an immune hypersensitivity to inflammatory stimuli [65,66]. The most common diseases in children are caused by viral infections, which sometimes progress to severe conditions with multiple organ failure (encephalopathy, cardiomyopathy, liver failure, kidney failure, or respiratory failure) because of cytokine storm. The brain has not detected the influenza virus in encephalopathy caused by an influenza virus, a representative disease of organ failure induced by a viral infection. Immune overreaction against a viral infection induces apoptosis due to cytokine storm. The pathology of aggravating viral infection is the immune overreaction to inflammation rather than direct damage of the virus [65,67,68,69]. Inflammatory stimulations can easily aggravate patients with autoinflammatory diseases because of the constitution of immune hypersensitivity [66]. A child’s immune system is the most sensitive, indicating that immune overreaction in children may be the most adverse effect of MIA. 

### 4.2. Immune Dysfunction Caused by Prenatal Exposure to LPS

LPS is used in MIA studies; however, these studies are less likely to be performed than studies using poly (I:C) as maternal inflammatory stimulation. The animal models of MIA induced by LPS are implicated in immune response modifications in offspring [70,71,72,73,74,75,76,77,78,79,80]. Previous studies have reported that cytokine levels at baseline were not changed by maternal LPS exposure in the prenatal period [70,72,75]. However, another study indicated high cytokine levels, including IL-1, IL-6, and IL-10, at baseline in offspring [71]. Exposure to maternal LPS in the prenatal period leads to different immune responses, such as immune overreaction or immune response failure, by the second attack of inflammation after birth (Figure 2) [70,71,72,74,75,79]. The animal model of immune overreaction shows high levels of cytokines, such as IL-1, IL-6, IL-17, and TNF-α, in serum after inflammatory stimuli. Contrarily, some reports also show that cytokine levels could not increase after inflammatory stimulation compared with control in offspring, and anti-inflammatory cytokines such as IL-10 showed a similar tendency [70,80] (Table 2). Immune response failure caused by exposure to MIA is implicated in suppressing MAPK p42/44 or delayed immune system maturation [70,75,80]. The reports on the mechanisms of immune dysfunction are scarce, and the mechanisms’ details are still unclear. The immune overreaction might be associated with exposure to maternal LPS from the first to second trimester or low dose. The immune response failure might be associated with exposure to maternal LPS in the third trimester or high doses. The immune responses vary depending on the characteristics of maternal inflammatory stimulation, such as gestation period, the dose of prenatal inflammatory exposure, or the time of sample collection after the second attack of inflammation in offspring.

## 5. The Gestation Period, Inflammatory Magnitude, Inflammatory Type of MIA, and Immune Dysfunction Mechanism in Offspring

### 5.1. Alteration of the Immune System Affected by the Time of MIA

Prenatal development of the immune system in rodents is almost similar to that of humans. The prenatal immune system in rodents comprises cells originating from primitive hematopoiesis in York sac at 7.5 days’ gestation. As the pregnancy progresses, the primary site of hematopoietic changes from the York sac to intraembryonic AGM, liver, bone marrow, thymus, and spleen. Diverse cell types in the immune system develop and mature at different gestational stages. Exposure to the maternal environment might significantly influence the fetal immune cells from the second to the third trimester because the critical period of immune system development in rodents is from 7.5 days’ gestation to birth [81,82]. Recent studies support this hypothesis. In a meta-analysis investigating the association between MIA and immunological disorders [83,84], exposure to MIA at midgestation causes immune overreaction from the prenatal period to the preweaning stage, and the period of maternal inflammatory stimulation is one of the essential factors of immune dysfunction after birth [61]. The risk of schizophrenia is seven-fold in early pregnancy and three-fold in mid-pregnancy during the influenza pandemic [85]. Furthermore, microglia, the macrophages of the central nervous system, are derived from primitive myeloid progenitors of mouse embryos at 8 days’ gestation and have been implicated in the pathogenesis of neurodevelopmental disorders [86]. The neurodevelopmental system is the most affected system at an early period of gestation. The effects of maternal infection on each organ and tissue might depend on the difference in the period when the inflammatory stimulation was received. Therefore, prenatal exposure to maternal infection is one of the essential factors of damage to organs and tissue, and a clinical phenotype is changed after birth.

### 5.2. Alteration of the Immune System Due to the Magnitude of Inflammatory Response or Type of Inflammatory Cytokines Present in the Prenatal Period

There are no reports about the importance of the magnitude or type of proinflammatory cytokines in abnormal immunity induced by MIA. Previous studies have shown an association between neurodevelopmental disorders and maternal influenza infection. Brown et al. examined the medical records of pregnant women and found an increased risk for schizophrenia caused by a maternal respiratory infection in offspring [81,85]. Neurodevelopmental disorders are possibly caused not only by influenza infection but also by other viral infections during gestation. These results indicate that the direct damage of the virus and immune regulatory components, such as cytokines or transcriptional factors, induced by MIA are crucial factors in organ injury or dysfunction after birth. Representative medicines, such as LPS or poly (I:C), used in MIA animal models activate different immune pathways. Their receptors include TLR-3 for poly (I:C) and TLR-4 for LPS. TLR-3 and TLR-4 activate NF-κB and AP-1 by the MyD88-dependent and Trif-dependent pathways, respectively, and result in the production of type I interferon and inflammatory cytokines [82,87]. Prenatal exposure to poly (I:C) or LPS leads to different neurodevelopmental or immune dysfunction disorders phenotypes.

Both poly (I:C) and LPS treatments cause anxiety-like behaviors in offspring. Poly (I:C) injection during gestation delays growth and sensorimotor development. LPS injection during gestation leads to reduced food intake and decreased body weight. IL-2, IL-5, and IL-6 serum levels in cases receiving poly (I:C) treatment are higher than in those receiving LPS treatment during gestation [88]. Moreover, the high concentrations of cytokines, such as IL-1β, IL-6, IL-8, IL-17, and IFN-γ, are linked to neurodevelopmental disorders in offspring [89,90,91,92]. Therefore, the difference in clinical features in neurodevelopmental disorders depends on the type of inflammatory molecules induced by the maternal inflammatory response in the fetus. In an animal model of schizophrenia, the increased cytokine level in maternal serum and fetal hippocampus induced by the injection of poly (I:C) was highly correlated with hippocampal neurogenesis impairment in offspring [93,94]. The magnitude of inflammatory response is strongly associated with the damaged tissue or organ phenotype. These results showed that the onset of neurodevelopmental disorders depends on the intensity of the inflammatory response and the kind of immunoregulatory molecules induced by inflammatory stimulation. Although an association between the mechanisms of immune dysfunction caused by MIA in offspring and the methods of inflammatory stimulation in pregnancy is partially elucidated, the whole picture of these mechanisms is still unclear. The elucidation of these findings is expected to contribute to the pathology of patients with immunological disorders, such as those with severe infections, autoinflammatory diseases, allergic diseases, and immunodeficiency diseases.

## 6. Epigenetic Changes in the Immune System

### 6.1. Importance of Epigenetic Alterations in Life

Epigenetic modifications, including DNA methylation, histone modification, and noncoding RNAs, often result in altered gene regulation without changing the DNA sequence and are associated with gene expression by changing the chromatin architecture [95,96]. Previous studies have reported that DNA methylation as well as hypomethylation can affect gene expression in various diseases and lead to schizophrenia by increasing transposon transfer [39,97]. Women in early pregnancy are more susceptible to these alterations [98,99,100], which could cause various diseases in offspring [101]. Moreover, prenatal exposure to famine could lead to epigenetic modifications, including low DNA methylation of the imprinted IGF2 gene; this epigenetic alteration persists six decades later in life [102,103]. Therefore, the epigenetic alterations due to exposure to adverse stimulations in fetuses affect them at birth and throughout their lifetime. The recent studies are focused on postnatal risk factors of each disease, which has slowly clarified the details of these factors. However, to understand the various diseases from the viewpoint implicated in the prenatal environment and epigenetic modifications, we can interpret the pathology of diseases more deeply. Research on the diagnosis and therapy by analyzing genes and the epigenome in the prenatal period is in progress. For example, several epigenetic abnormalities have been identified in patients with tumors, autoimmune diseases, diabetes mellitus, hematologic diseases, neurodevelopmental disorders, and infections [104].

### 6.2. Epigenetic Changes Induced by MIA in the Immune System

There are no reports on the direct relationship between epigenetic changes of immunological disorders in offspring and MIA; however, environmental factors other than MIA cause epigenetic changes in the immune system. Exposure to maternal farm environments increases the number of T regulatory cells in infants and decreases Th2 cytokine levels. These alterations are associated with demethylation at the forkhead box P3 promoter, which is one of the main transcription factors of Treg and is implicated in the immune response to inflammatory stimulation [105]. In animal models, immune molecules, such as cytokines and transcriptional factors, are transported from the dam to the pup through the placenta and change the activation of epigenetic modification enzymes. These modifications of epigenetic enzymes alter gene regulation of cytokine or transcriptional factors in the immune cells or signaling pathway [106,107,108,109]. IL-6 can enhance the activation of DNA methyltransferase 1 (DNMT1), suggesting the direct relationship between MIA and epigenetic alteration [107]. IL-17 may inhibit the histone deacetylase (HDAC) activity through the PI-3Kinase signaling pathway [108]. The main signaling pathways associated with the immune response, activated by cytokines, are JAK/STAT and MAPK/ERK signaling pathways. STAT proteins regulate histone acetylation at STAT-binding areas [106]. The MAPK/ERK signaling pathway is associated with epigenetic regulators involved in the histone acetyltransferase to chromatin [109]. HDAC induces an epigenetic modification of IL-10 expression [110]. H3K4 methylation suppresses NF-κB and results in decreased IL-6 expression [111]. Other studies show epigenetic modifications, such as DNA methylation, acetylation associated with polarization from naive T cells to Th1 and Th2, or cytokine production by a helper T cell [112,113,114]. Epigenetic alterations in the immune system cause various adverse effects on the host defense and may induce immune overreaction or immune response failure because of alterations in the gene expressions of inflammatory cytokine or transcriptional factors by changing the chromatin structure [115,116]. Further studies are required to investigate whether MIA directly interacts with epigenetic modifications of the immune system in offspring.

### 6.3. Prospects of Prevention and Treatment Using Epigenetic Therapy in the Prenatal Period

Prenatal diagnosis and treatment focus on preventing the onset of various diseases with genetic or epigenetic factors and reducing the medical costs for patients. Gene therapy for autoimmune disease in the postnatal period regulates the immune system, including the levels of proinflammatory cytokine or immune molecules or infiltration of lymphocytes. However, animal models of prenatal gene therapy for immunological disorders have yet to be reported. Gene therapy of an injection vector in the prenatal period can prevent hemophilia in offspring and is expected to be applied as a prenatal treatment for various diseases [117,118]. Epigenetic therapy in the postnatal period could treat diseases such as myelodysplastic syndrome, leukemia, cancer, heart failure, and diabetic retinopathy [104]; however, there are no reports about epigenetic therapy in the prenatal period. The application of epigenetic therapy during the prenatal period may suppress the onset of fatal diseases in the prenatal or neonatal period. However, this therapy has some disadvantages; it is not indicated for target cells, and activating gene expression in normal cells might lead to cancer [104]. We need to overcome many problems in the clinical field; however, significant benefits await epigenetic therapy’s development. Thus, we need to evaluate the relationship between epigenetic alteration, maternal environments, and postnatal effect using an animal model.

## 7. Conclusions

### The Association of Genetic Factors, Maternal Infection, and Postnatal Environmental Factors in Immune Dysfunction

Genetic factors are crucial to the onset of several human diseases, which are directly related to genetic diseases and are indirectly implicated in many diseases. In epidemiologic studies, exposure to a viral infection is the most critical factor that excessively activates the immune response in offspring. Maternal bacterial infection causes different immune responses in offspring, including immune overreaction or immune response failure. These clinical features in the immune system may depend on time, magnitude, and inflammatory type of MIA. In addition, postnatal environments are the ultimate determinants of immune overreaction or immune response failure. Immune overreaction leads to various diseases with immune hypersensitivity, including autoinflammatory, autoimmune and allergic diseases, and severe viral and bacterial infections. Immune response failure in offspring might cause immunodeficiency diseases and suppression of diseases with immune hypersensitivities, such as autoinflammatory, autoimmune, and allergic diseases and severe infections [119] (Figure 3).

The goal of therapies was mainly to cure the disease or disorder after its onset until a few decades ago. Recently, treatments have been performed to prevent various contracting diseases. For instance, the treatment for hypertension, obesity, hyperlipidemia, and diabetes mellitus is aimed at preventing coronary artery disease or stroke in old age. Medical treatment is essential to prevent the crisis by removal of the risk factor. Additionally, the development of epigenetic alterations and genetic analysis at birth or shortly after birth may allow for the identification of individual disease susceptibility. In the future, the diagnosis and treatment of these diseases in the prenatal period may be achieved using an epigenetic analysis and suppression of various diseases associated with epigenetic modifications.

The determinants of the development of immune disorders, such as immune overreaction and immune response failure, in offspring depend on the characteristics of maternal inflammatory stimulation, including gestation period, the magnitude of maternal inflammation, and the kinds of immune molecules induced by MIA. The genetic factor and prenatal stimulations lead to the development of various diseases associated with immune system disorders. These constitutions are exacerbated by exposure to postnatal risk factors in offspring, leading to various diseases with abnormal immunity.

## Figures and Tables

**Figure 1 cells-12-00741-f001:**
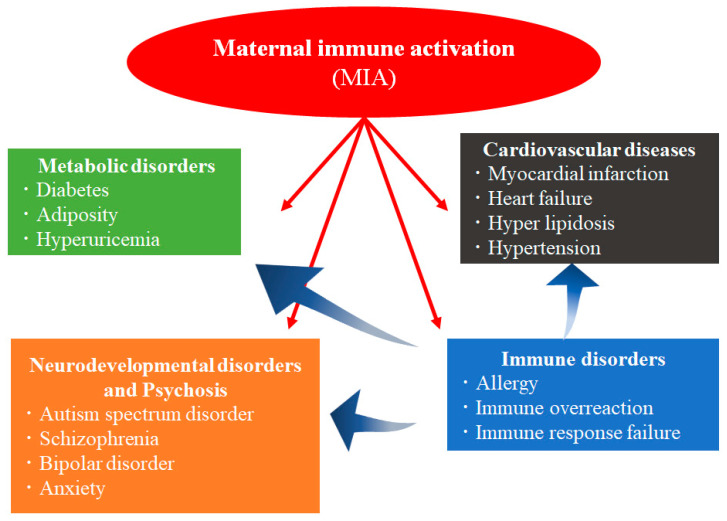
Effects of exposure to MIA in offspring. Exposure to MIA in the prenatal period induces various diseases, such as metabolic, neurodevelopmental, immune system disorders, and cardiovascular diseases. MIA is one of the risk factors for developing these diseases and disorders in the postnatal period. Immune system disorders might cause these diseases due to MIA exposure in the prenatal period.

**Figure 2 cells-12-00741-f002:**
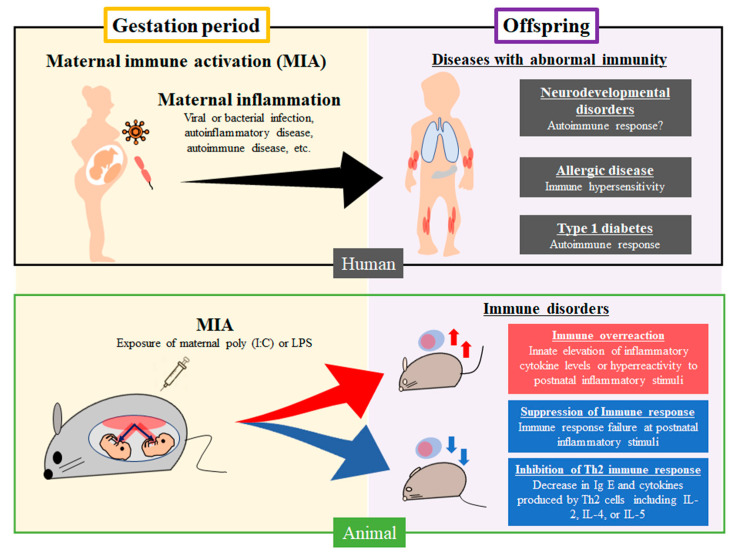
Epidemiologic and experimental data associated with MIA and immune disorders in human and animal models.

**Figure 3 cells-12-00741-f003:**
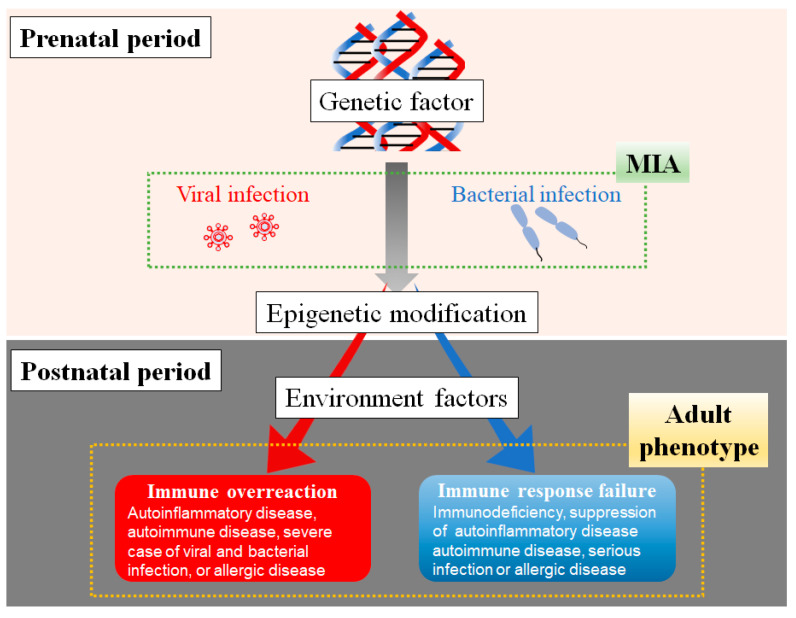
The association of genetic factors, maternal immune activation (MIA), and environmental factors in immune dysfunction after birth.

**Table 1 cells-12-00741-t001:** Association between MIA caused by poly (I:C) and immune disorders caused by postnatal inflammatory stimuli.

Literature Authors (Year) (Ref#)	Species	Treatment of Pregnant Dam		Postnatal Treatment of Offspring				
		First Stimulation (mg/kg BW)	Period (Gestational Day, GD)	Second Stimulation	Period (Postnatal Day, PD)	Findings	Histopathology	Assumed Pathogenesis
Charity et al. (2014) [68]	Mouse	Poly (I:C) (20)	Third trimester (GD12.5)	LPS and IFN-γ	PD 21	Increase in IL-1, and IL12 in vitro	NE	Macrophage 1 polarization
Destanie et al. (2017) [67]	Monkey	Poly (I:C) (2500)	First and second trimester (GD 43,44, 46, 100,101, and 103)	LPS or Poly (I:C)	1 year	Increase in IL-1, IL-2, IL4, IL6, IL12, and TNF-α in vitro	NE	Activation of innate and Th2 immune response
Shimizu et al. (2021) [66]	Mouse	Poly (I:C) (20)	Third trimester (GD 12.5, 14.5, and 16.5)	Poly (I:C)	PD 21-28	Increase in IL-6, IL-17, and TNF-α in serum	Liver necrosis	Unfolded protein response defects

MIA, maternal immune activation; poly (I:C), polyriboinosinic-polyribocytidylic acid; LPS, lipopolysaccharide, NE: not examined.

**Table 2 cells-12-00741-t002:** Association between MIA using LPS and immune disorders caused by postnatal inflammatory stimuli.

Literature Authors (Year) (Ref#)	Species	Treatment of Pregnant Dam		Postnatal Treatment of Offspring				
		First Stimulation (μg/kg/dose)	Period (Gestational Day, GD)	Second Stimulation	Period (Postnatal Day, PD)	Findings	Histopathology	Assumed Pathogenesis
Lasaka et al. (2007) [83]	Rat	LPS (500)	Third trimester (GD 18)	LPS	PD 21	Decrease in IL-1, IL-6, and TNF-α in serum	NE	ND
Surriga et al. (2009) [73]	Rat	LPS (500)	Third trimester (GD 18)	LPS	PD 21	Decrease in IL6 mRNA expression in the liver	NE	Suppression of MAPK P42/44
Basta-Kaim et al. (2012) [77]	Rat	LPS (1000)	Second to third trimester (Every 2days from GD 7)	Concanavalin A	PD 30 and 90	Increase in IL-1β, IL-2, IL-6, and TNF-α in vitro	NE	Increased proliferative activity of splenocytes
Kirsten et al. (2013) [75]	Rat	LPS (100)	Second trimester (GD 9)	LPS	PD 60-67	Increase in IL-1β in serum	NE	Glucocorticoid dysregulation
Zager et al. (2013) [85]	Mouse	LPS (120)	Third trimester (GD 17)	LPS	PD 70	Increase in IL-12 in vitro	NE	Skewing of the cytokine balance towards Th1
Hsueh et al. (2017) [75]	Mouse	LPS (25, 25, 50)	Third trimester (GD 15, 16 and 17)	LPS	PD 56	Increase in IL-1, IL-6, IL-10, IL-12, IL-17, TNF-α, and IFN-γ in serum	NE	Increase in MCP-1 level
Adams et al. (2020) [82]	Mouse	LPS (10)	First to third trimester (GD 0, 7, 14)	LPS	PD 49	Increase in IL-1, IL-6, and IL-10 mRNA expression in the spleen	NE	Glucocorticoid dysregulation

MIA, maternal immune activation; LPS, lipopolysaccharide, NE: not examined, ND: not determined.

## Data Availability

Not applicable.

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
