# Peer review of "Influence of Immune System Abnormalities Caused by Maternal Immune Activation in the Postnatal Period"

_cells, 2023, doi:10.3390/cells12050741_

Round 1

Reviewer 1 Report

The review touches an important topic and is very timely given the concerns raised by developmental immunologists regarding the exposure of millions of fetuses to maternal inflammation during the COVID-19 pandemic. Throughout the article, the authors repeat general statements regarding the effects of MIA on fetal immune activation, immune suppression and/or epigenetic changes but do not provide information beyond that.  The authors repeat statements like “After birth, maternal inflammation leads to abnormal immunity, such as immune overreaction or autoimmunity” but these are not followed by more detailed information to enlighten the reader about specific pathways and immunological mechanisms. For example, the authors explain the immune regulatory effects of maternal farm exposure with innate immune activation by maternal LPS. However, they do not provide any details on why or how maternal farm contact leads to maternal LPS exposure and why this interaction is protective against offspring allergic diseases. Therefore, the article could be significantly shortened without losing content. On the plus side, the illustrations are clear and provide a good summary of what is being mentioned in the article.

Author Response

Responses to reviewer’s comments

HANDLING EDITOR

Thank you for your constructive comments that strengthen the manuscript.

The reviewers' concerns have been fully addressed and the manuscript has been revised.

The revised text is also highlighted in the manuscript.

We hope that this manuscript will be published in Cells.

RESPONSES TO REVIEWER #1

We appreciate your valuable suggestions for improving the manuscript

1) We have added the detailed information implicated with immune system and cell stress response induced by MIA in prenatal period. Moreover, we described the protective mechanism against allergic diseases in offspring because of maternal farm contact as below;

"The frequency of contact with maternal farm environments during gestation is inversely correlated with the incidence of allergic diseases in offspring. Contact with live-stock or drinking fresh milk can result in exposure to multiple bacteria, endotoxin, and fungi. Immune responses in newborns with a history of maternal farm contact were weaker than in newborns in the non-maternal farm contact group. Interferon (IFN)-γ/IL-13 ratio increased after LPS stimulation in farming mother group, and immune response tends to be Th1 dominant. Moreover, the number of Th9 cell has been shown to increase in allergic patients. IL-9 production decreased after LPS stimulation in cord blood mononuclear cells from neonates exposed by maternal farm contact and might be implicated with suppression of Th9 function. The quality and function of neonatal Treg induced immune response imbalance of Th1 dominant and suppressed the function of Th9 cell and the production of IL-9 [23].”

2) As one of the suggestions from Reviewer 1 regarding the association between MIA exposure and the risk of developing disorders in Offspring, the revised manuscript discussed the involvement of UPR as a cause of the increased risk of developing organ disorders due to MIA exposure, referring to the relevant literature as follows.

"The mRNA expression of binding immunoglobulin protein (Bip) and activating transcription factor 4 (Atf4), one of the major regulators and key transcription factors of the unfolded stress response (UPR), was low compared to controls, suggesting that the UPR defect is induced by prenatal MIA exposure.
UPR defects result in the inability to maintain tissue and organ homeostasis due to excess unfolding proteins, leading to excessive activation of the immune response to inflammatory stimuli [58]."

3) We have also improved the manuscript by removing duplicate contexts, phrases, and discussions to avoid redundant descriptions.

With this revision, we hope our manuscript is suitable for publication in Cells.

Reviewer 2 Report

The reviewed article is devoted exclusively to the urgent problem of studying the activation of the maternal immune system as factor causing fetal development disorders, the consequences of which can lead to various diseases in the future. The review summarizes the current information confirming the hypothesis of the developmental origins of health and disease. Of particular interest is the data devoted to the consequences that covid-19 leads to during pregnancy. The authors consider in detail the issues concerning the development in the prenatal and early postnatal period of immune disorders arising from the activation of the maternal immune system (MIA), accompanied by increased production of proinflammatory cytokines and leading to various metabolic, cardiovascular, neurodevelopmental diseases and psychosis. Of particular interest is the part devoted to the consequences that Covid-19 leads to during pregnancy. It is very important that the authors discuss the role of epigenetic changes induced by MIA in the immune system, since the epigenetic posttranslational mechanisms associated with developmental disorders in early ontogenesis do not seem to be definitively clear and this problem is being actively discussed in the literature at present time.

As comments, it should be noted:

1. It is desirable to provide data on the possibility of changing the production of anti-inflammatory cytokines in MIA in the postnatal period and their contribution to this process.

2. As factors activating the maternal immune system during pregnancy, it is advisable to consider such common effects as alcoholism and prenatal hyperhomocysteinemia.

3. It is hardly advisable to include the Strategy Search subsection in the article.

Author Response

Responses to reviewer’s comments 

HANDLING EDITOR

Thank you for your constructive comments that strengthen the manuscript.

The reviewers' concerns have been fully addressed and the manuscript has been revised.

The revised text is also highlighted in the manuscript.

We hope that this manuscript will be published in Cells.

RESPONSES TO REVIEWER #2

We agree with the reviewer’s suggestion and modified the manuscript. With this additional information, we hope our manuscript is suitable for publication in Cells.

The point-to-point responses have been addressed below, and the revised text is highlighted in the manuscript.

  1. It is desirable to provide data on the possibility of changing the production of anti-inflammatory cytokines in MIA in the postnatal period and their contribution to this process.

Response: We appreciate your valuable suggestions for improving the manuscript. MIA caused by poly (I:C) could not affect anti-inflammatory cytokine such as IL-4 or IL-10. MIA caused by LPS leads to various alterations of anti-inflammatory cytokines including increasing, decreasing, or no change. In patients with autism-like disorders, a typical neurodevelopmental disorder, there is no change in serum IL-10 levels in the offspring. A decrease in anti-inflammatory cytokines may lead to a decrease in anti-inflammatory effects and induce the development of various diseases involving immune overreaction (P9, Line256 and P10, Line 311).

  1. As factors activating the maternal immune system during pregnancy, it is advisable to consider such common effects as alcoholism and prenatal hyperhomocysteinemia.

Response: We agree with the reviewer's suggestion. We add that maternal alcoholic and hyperhomocystemia induce maternal immune activation in prenatal period. (P2, Line48)

  1. It is hardly advisable to include the Strategy Search subsection in the article.

Response: We agree with the reviewer's suggestion and have deleted search strategy paragraph. (P2, Line77)

Round 2

Reviewer 1 Report

Thank you for the edits. Unfortunately the grammar in the added sections of the manuscript makes the text difficult to read. For example "...the maternal immune response is activated by maternal infection, hyperhomocystemia, or alcoholic." Alcoholic what? The correct term is "hyperhomocysteinemia".

Another example is the sentence "The quality and function of neonatal Treg induced immune response imbalance of Th1 dominant and suppressed the function of Th9 cell and the production of IL-9 [23]." This sentence does not make any sense. 

Author Response

We express our heartfelt appreciation to the reviewers for their insightful comments on our manuscript. We hereby resubmit the manuscript to “Cells” after carefully considering and addressing the suggestions made by the reviewers. Please find the following “Responses to the reviewer’s comments.”

We appreciate the insightful suggestions of the reviewers, as their comments helped us revise and substantially improve the manuscript. We hope you will find this manuscript suitable for publication in Cells.

Sincerely,

Toshihisa Hatta, M. D., Ph. D.

Department of Anatomy, Kanazawa Medical University, 1-1 Daigaku, Uchinada, Ishikawa 920-0922, Japan

Tel: +81-76-218-8113

Fax: +81-76-218-8189

E-mail: thatta@kanazawa-med.ac.jp

Responses to reviewer’s comments

HANDLING EDITOR

Thank you for your constructive comments that strengthen the manuscript.

The reviewers' concerns have been fully addressed and the manuscript has been revised.

The revised text is also highlighted in the manuscript.

We hope that this manuscript will be published in Cells.

RESPONSES TO REVIEWER #1

We agree with the reviewer’s suggestion and modified the manuscript. With this additional information, we hope our manuscript is suitable for publication in Cells.

The point-to-point responses have been addressed below, and the revised text is highlighted in the manuscript.

  1. Thank you for the edits. Unfortunately, the grammar in the added sections of the manuscript makes the text difficult to read. For example, "...the maternal immune response is activated by maternal infection, hyperhomocystemia, or alcoholic." Alcoholic what? The correct term is "hyperhomocysteinemia".

Response: We agree with the reviewer's suggestion, and have revised the sentence as below; " It is a phenomenon in which maternal infections, autoimmune diseases, allergies, asthma, atherosclerosis, malignancy, hyperhomocysteinemia, and alcohol consumption activates maternal immune response" (P1, Line44).

  1. Another example is the sentence "The quality and function of neonatal Treg induced immune response imbalance of Th1 dominant and suppressed the function of Th9 cell and the production of IL-9 [23]." This sentence does not make any sense.

Response: We agree with the reviewer's suggestion, and have revised the sentence as below; "Alterations in the quality and function of neonatal Tregs induced an imbalance in the immune response, causing a Th1-dominant state and suppressing Th9 cell function and IL-9 production [23]" (P4, Line128).

 The English of the revised manuscript was proofread by an expert. With this revision, we hope our manuscript is suitable for publication in Cells.

Round 3

Reviewer 1 Report

Thank you for revising.